# Amniodrainage-Induced Circulatory Dysfunction in Women Treated for Twin-To-Twin Transfusion Syndrome

**DOI:** 10.3390/jcm9072085

**Published:** 2020-07-02

**Authors:** Patrick Greimel, Philipp Klaritsch, Holger Simonis, Bence Csapó, Maximilian Pohl, Daniel Schneditz

**Affiliations:** 1Department of Obstetrics and Gynecology, Medical University of Graz, Auenbruggerplatz 14, 8036 Graz, Austria; patrick.greimel@medunigraz.at (P.G.); bence.csapo@medunigraz.at (B.C.); 2Department of Anesthesiology and Intensive Care Medicine, Medical University of Graz, Auenbruggerplatz 14, 8036 Graz, Austria; holger.simonis@medunigraz.at; 3Otto Loewi Research Center, Medical University of Graz, Auenbruggerplatz 14, 8036 Graz, Austria; maximilian.pohl@stud.medunigraz.at (M.P.); daniel.schneditz@medunigraz.at (D.S.)

**Keywords:** twin-to-twin transfusion syndrome, fetoscopy, amniodrainage, hemodilution, intrauterine pressure

## Abstract

Twin-to-twin transfusion syndrome (TTTS) in monochorionic-diamniotic twin pregnancies usually requires fetoscopic laser ablation (FLA) followed by amniodrainage (AD). Perioperative maternal hemodynamic changes and hemodilution have been observed. Little is known about the underlying pathophysiology. We aimed to evaluate the impact of high volume amniodrainage on intrauterine pressure, placental thickness and maternal blood characteristics. A total of 18 cases of TTTS were included in this prospective pilot study. All patients were treated with FLA and subsequent AD. Intrauterine pressure and placental thickness were assessed before, during and after amniodrainage. Maternal hemoglobin, hematocrit and serum albumin were measured at admission and 24 h after the intervention. Amniodrainage led to a decrease in mean intrauterine pressure (from 30.1 ± 8.1 mmHg to 17.6 ± 3.6 mmHg (*p* < 0.001)) and an increase in mean placental thickness (from 16.8 ± 6.4 mm to 31.83 ± 8.64 mm (*p* < 0.001)). There was a positive correlation between changes in placental thickness and the amount of amniodrainage during intervention (Pearson’s Rho 0.73; *p* = 0.001). Hematocrit decreased from 33.4 ± 3.8 (%) to 28.4 ± 3.5 (%), i.e., an increase in relative blood volume by 18 ± 10.2% (*p* < 0.001). Albumin decreased from 37.9 ± 0.9 g/L to 30.7 ± 2.2 g/L, i.e., an increase in relative plasma volume by 24 ± 8.1% (*p* < 0.001). Amniodrainage leads to uterine decompression, increased placental thickness and subsequent maternal hemodilution. We propose the term “amniodrainage-induced circulatory dysfunction” for these specific maternal hemodynamic changes in the treatment of twin-to-twin transfusion syndrome.

## 1. Introduction

Twin-to-twin transfusion syndrome (TTTS) complicates approximately 10% of monochorionic-diamniotic (MCDA) twin pregnancies [1]. Induced by unbalanced intertwin blood flow through vascular anastomoses on the placental surface, the condition leads to volume depletion in the donor twin and simultaneous volume overload in the recipient twin. Left untreated, the process is mostly progredient and in these cases fetal outcome is usually unfavorable [2]. Fetoscopic laser ablation (FLA) of placental anastomoses is currently the best treatment available for TTTS, yielding double survival rates of more than 60% in large recent series. FLA is usually completed by amniodrainage of the recipient´s excess fluid between 1000 and 4000 mL to achieve normal amniotic fluid levels [3,4].

In contrast to the high number of scientific reports on fetal outcome and treatment options, data on maternal hemodynamic changes or adverse side effects following FLA procedures are scarce. Clinical signs suggestive of maternal hemodilution were described in cases of mid-trimester TTTS managed by FLA and subsequent amniodrainage, providing evidence that such interventions have a significant impact on the maternal compartment [5,6,7]. A variety of observations including maternal hemodynamic adaptations, decrease in intrauterine pressure and increase in placental volume have been described following high volume amniodrainage [8,9]. The observed hemodynamic characteristics may be severe and are then akin to the so-called Ballantyne syndrome or “mirror syndrome” [10]. However, the mechanisms leading to these complications are still not comprehensively understood.

In a previous retrospective study, we observed that high volume amniodrainage was significantly correlated with maternal hemodilution and hemodynamic alterations [5]. Thus, we hypothesized that the frequent observation of hemodilution is caused by high volume amniodrainage after FLA procedures. We therefore sought to prospectively investigate the impact of FLA procedures and high volume amniodrainage on intrauterine pressure decrease, subsequent placental thickness gain and maternal blood characteristics.

## 2. Materials and Methods

A total of 18 women at a minimum age of 18 years with MCDA twin pregnancies complicated by TTTS were recruited for this prospective pilot study at the Department of Obstetrics and Gynecology at the Medical University of Graz, Austria, between November 2017 and September 2019. Fetoscopic laser ablation of placental anastomoses was performed between 16 and 26 weeks of gestation following international recommendations [11]. In one case, we performed FLA after 26 weeks due to rapidly progressing disease on the explicit request of the parents [12].

At our institution, intrauterine treatment by FLA is generally offered to all cases with Quintero stage I or higher, as defined by polyhydramnios in the recipient‘s sac (i.e., a deepest vertical pocket (DVP) measuring more than 8 cm before 20 weeks of gestation and more than 10 cm beyond 20 weeks of gestation) in combination with oligohydramnios in the donor‘s sac (i.e., a DVP of less than 2 cm) [13].

FLA was performed in a standardized fashion as previously described [5]. All FLA procedures were completed by amniodrainage from the recipient‘s amniotic sac until a DVP of about 6 cm was achieved. Amniodrainage was carried out via a 10 Fr cannula (Cook Medical, Bloomington, IN, USA) and the drained volume was measured using a graduated beaker. The procedure was constantly monitored by ultrasonography. In general, there was no relevant perioperative fluid administration except intravenous cefazolin (3 g every 8 h for 24 h) and continuous intravenous administration of remifentanil (0.1–0.2 µg/kg/min) during the intervention. Two hours after the intervention, the women were allowed unrestricted fluid and food intake.

Intrauterine pressure was assessed by a TruWave digital pressure sensor (Edwards Lifescience LLC, Irvine, USA) in 10 consecutive cases. The pressure sensor was directly linked to the aforementioned cannula via a Luer lock connection and a flexible sterile tube without requiring an additional incision for measurements. The sensor was zeroed to the level of the mid-axillary line of the pregnant woman placed in supine position. Intrauterine pressure was recorded under “hands-off” and “stop-flow” conditions before and during amniodrainage. Measurements were conducted in steps of 200 mL of amniodrainage. Pressures obtained during uterine contractions were discarded. Data were digitally processed and stored on a Datex-Ohmeda unit (GE Health Care, Chicago, IL, USA) and recorded in written form as well.

We used a standardized approach for the evaluation of the placental thickness. The same expert sonographer conducted all measurements. Placental thickness was defined as the maximum distance from the myometrium–placenta interface to the placenta–amniotic fluid interface. Measurements (n = 18) were performed on an adequately sized and focused ultrasound image in an anterior–posterior view on a Voluson E10 or E6 (GE Health Care, Chicago, IL, USA) under the same conditions as the intrauterine pressure measurements. Maximum thickness was obtained in a linear fashion and taken from the very same site of the organ at all timepoints. To ensure measurements from identical placental sites, we linked the position of the ultrasound probe to the site of umbilical cord insertion or to major maternal abdominal vessels in case of posterior wall location of the placenta [14,15].

Maternal age, gestational age at intervention, maternal body mass index (BMI) and Quintero stage at time of intervention were documented. The total volume of amniodrainage was recorded and maternal hemoglobin (Hb), hematocrit (Hct) and serum albumin were measured at admission (pre) and 24 h after the intervention (post) by standard laboratory techniques. Relative hemodilution for blood (DB) and plasma compartments (DP) were quantified as the ratios of “post to pre” concentrations. The relative expansion of blood (RBV) and plasma volumes (RPV) by hemodilution was measured as the ratio of “pre to post” concentrations as described elsewhere [16].

Data were tested for normal distribution by the Kolmogorov–Smirnov test and the Shapiro–Wilk test. Non-parametric data were tested with the Wilcoxon test. In case of normal distribution, a paired t-test was performed. A probability of < 0.05 was considered statistically significant. Analyses were carried out using SPSS Statistics 25 (IBM, Armonk, NY, USA).

The current study was approved by the Ethics Committee of the Medical University of Graz, Austria, registered at Institutional Review Board Registry as IRB00002556 (30-035 ex 17/18). Written informed consent was obtained from all patients before intervention. All patient data were pseudo-anonymized. Only authorized persons had access to the original data sets.

## 3. Results

The study included 18 women with TTTS treated by FLA and subsequent amniodrainage. Median maternal age at intervention was 28 years (21–39) and maternal BMI was 25.9 (17.4–38.9). Gestational age at intervention was 21.1 (19.3–28.1) weeks (Table 1). There were six cases of Quintero stage I (33.3%), five cases of stage II (27.8%) and seven cases of stage III (39.9%). The median volume of amniodrainage at intervention was 1675 mL (1000–4000 mL).

Intrauterine pressure decreased and placental thickness increased in every study. There was a significant decrease in mean intrauterine pressure from 30.1 ± 8.1 mmHg before amniodrainage to 17.6 ± 3.6 mmHg after amniodrainage (*p* < 0.001) (Figure 1), corresponding to a mean percentual drop of 41 ± 7.7 % from baseline measurements (*p* < 0.001). Mean placental thickness increased from 16.8 ± 6.4 mm before amniodrainage to 31.8 ± 8.6 mm after amniodrainage (*p* < 0.001) (Figure 2), corresponding to a mean percentual increase of 105 ± 47.0 % from baseline measurements (*p* < 0.001) (Table 2).

We found a significant correlation between the volume of amniodrainage and the effects on placental thickness. From pre- to post-drainage, there was a positive correlation between changes in placental thickness and the amount of amniodrainage during intervention (Pearson´s Rho 0.73; *p* = 0.001).

Maternal blood characteristics revealed significant changes between pre- and postoperative measurements (*p* < 0.001 for all parameters) (Table 2 and Table 3). Mean hemoglobin (Hb) dropped from 112.1 ± 12.0 g/L to 95.1 ± 12.1 g/L and mean hematocrit (Hct) from 33.4 ± 3.8 (%) to 28.4 ± 3.5 (%), corresponding to a 1.2 ± 0.1-fold relative expansion of blood volume (RBV), i.e., an average increase in blood volume by 18.0 ± 10.2 %. In seven cases, data on albumin were also available: mean albumin dropped from 37.9 ± 0.9 g/L to 30.7 ± 2.2 g/L, corresponding to a relative expansion of plasma volume (RPV) of 1.2 ± 0.1, i.e., an average increase in plasma volume by 24 ± 8.1 %. The mean increase in RBV was 18.0 ± 10.2 % in the whole study population (n = 18), whereas the mean increase in RBV was 18.0 ± 7.0 % in the group in which serum albumin was also available. Since these two values (SD) are similar in both groups, we consider the observed maternal plasma volume changes (RPV), which are based on serum albumin measurements of the smaller group (n = 7), representative of the whole study population.

## 4. Discussion

In this prospective pilot study, we observed a significant increase in placental thickness while intrauterine pressure decreased significantly during high volume amniodrainage. Postoperative maternal hemodilution was found in every study and was linked to these utero-placental changes.

Severe polyhydramnios results in increased intrauterine pressure in cases with TTTS [17], while amniodrainage leads to a pressure decrease [9,18,19,20]. In the current study, we were able to quantify those findings by using a digital high-sensitive pressure assessment during ten consecutive fetoscopic interventions. In contrast to the majority of prior reports, our data were not derived from serial amniodrainage for TTTS but from intraoperative measurements before and during amniodrainage in the recipient’s amniotic cavity immediately after fetoscopic laser ablations. Baseline pressures before amniodrainage were significantly elevated in comparison to data on the intrauterine pressure of healthy controls [21,22]. This indicates a state of intrauterine hypertension during the presence of severe TTTS establishing a type of “intrauterine compartment syndrome” and resembling abdominal compartment syndrome [23]. Therefore, amniodrainage seems to be a crucial step in the treatment of TTTS in combination with FLA. Such decompression of the intrauterine compartment changes feto–placental as well as maternal–placental perfusion, which has already been demonstrated by several study groups [24,25,26]. However, uterine decompression might also have negative implications for fetuses and pregnant women. High-volume amniodrainage lowers intrauterine pressure and in return leads to a relevant increase in placental thickness. This seems to cause transient hemodynamic changes in both the fetal and the maternal compartments, comparable to hemorrhage. Depletion of effective circulating volume in the fetuses may impair fetal brain perfusion and add to the risk of adverse neurodevelopmental outcome, especially when amniodrainage is performed without preceding FLA [11,27].

In a previous study, we observed that high volume amniodrainage was significantly correlated with maternal hemodilution and hemodynamic alterations [5]. We could confirm those findings in the current prospective cohort, by demonstrating a drop in hemoglobin concentration and hematocrit in the absence of clinical blood loss, suggesting an expansion of blood volume by 18.0 ± 7.0 % (Hct). The drop in plasma albumin was even larger, indicating an increase in plasma volume of 24.0 ± 8.1 %. This difference can be explained by the relationship between plasma and blood volumes as determined by hematocrit and analyzed elsewhere [28]. The albumin concentration predicted from hematocrit changes (30.0 g/L ± 3.3 g/L) was not different from the measured albumin concentration (30.7 g/L ± 2.2 g/L) (*p* = 0.63). This is an indirect confirmation that the hematocrit dropped as a result of hemodilution and not because of blood loss.

A decrease in intra-uterine and intra-abdominal pressures leads to an increase in the transmural pressure in the vascular system of the uterus and the intra-abdominal vasculature, because the volume within arteries and veins is a function of both the pressure difference across their wall, the transmural pressure, and their distensibility, the vascular compliance [29]. Since vascular areas involved in amniodrainage are characterized by high compliance [30], the increase in transmural pressure will also increase the intravascular blood volume. The resulting sequestration of blood in compliant vascular beds of uterine and abdominal circulations therefore resembles an acute hemorrhage. As with hemorrhage, intravascular volume loss is refilled from the extravascular space to compensate for the effective volume deficit.

The following factors are likely to contribute to the observed circulatory findings in the maternal compartment:

Firstly, uterine decompression and the subsequent gain of placental thickness increases maternal plasma volume on the basis of modified placental blood rheology and a reduction in effective microfiltration pressure due to the loss of vascular resistance [31,32]. The acutely dilated placental vascular system and the intervillous space, that essentially is a large lake of blood exerting little resistance to blood flow, is likely to reabsorb more fluid than it loses through filtration. Any reduction in resistance will increase flow, and potentially lead to further vasodilatation through increased shear stress and nitric oxide mediated pathways in a feed-forward fashion [30]. Data on a reduced pulsatility index (PI) of the middle cerebral artery after therapeutic amniodrainage in twin gestations gathered by Mari et al. support the idea of a reduced effective blood volume of the feto–placental unit [24,27].

Secondly, amniodrainage reduces the physical size of the pregnant uterus. Thus, mechanical decompression of abdominal arterioles and venules may cause splanchnic vasodilatation and increased filling of the vascular bed, comparable to hemodynamic changes observed with large volume paracentesis of tense ascites in hepato-renal syndrome and portal hypertension, also known as paracentesis-induced circulatory dysfunction (PICD) [33].

Thirdly, reduction in uterine size due to amniodrainage may decrease the intraabdominal pressure and therefore increase the venous backflow, especially from the lower extremities of the patient that are placed in a horizontal position during the intervention and during the 24 h observation period after intervention. This might mobilize fluid from the peripheral edema of the lower extremities, i.e., extracellular fluid volume, into the intravascular space, comparable to volume shifts observed during peritoneal dialysis [34].

Although we provided some possible mechanisms of the present phenomenon, we did not formally examine if these really do occur in our patients. Our explanations should provide a common basis for further investigations and an invitation for scientific discussion and co-operation.

The positive correlation between the volume of amniodrainage and the increase in placental thickness supports the hypothesis that uterine decompression might provoke placentomegaly resembling placental hydrops. Placental hydrops is frequently reported to occur in the course of Ballantyne´s Syndrome, usually referred to as “mirror syndrome”. In our study, we demonstrated that uterine decompression is linked with the rapid increase in placental thickness and subsequent hemodilution in the very same patient group. This supports the hypothesis that high-volume amniodrainage might cause a subclinical and mild variant of iatrogenic Ballantyne´s Syndrome. To clarify the terminology, we suggest the term amniodrainage-induced circulatory dysfunction (AICD) for this attenuated variant of iatrogenic Ballantyne´s Syndrome in line with findings made during paracentesis for cirrhotic ascites [35].

The study population and patient characteristics were comparable to previous publications on this topic [7,8,18,19,20]. However, the limitations of a pilot study with a small number of cases apply to our study. Furthermore, we did not stratify for gestational age which might be a limitation of this study due to continually expanding plasma volume during physiological pregnancy [36]. The strength of this study is the holistic approach to maternal monitoring during intrauterine interventions.

## 5. Conclusions

Amniodrainage is linked with uterine decompression, increased placental thickness and subsequent maternal hemodilution. We propose the term “amniodrainage-induced circulatory dysfunction” for specific maternal hemodynamic changes observed after iatrogenic uterine decompression in the treatment of TTTS.

## Figures and Tables

**Figure 1 jcm-09-02085-f001:**
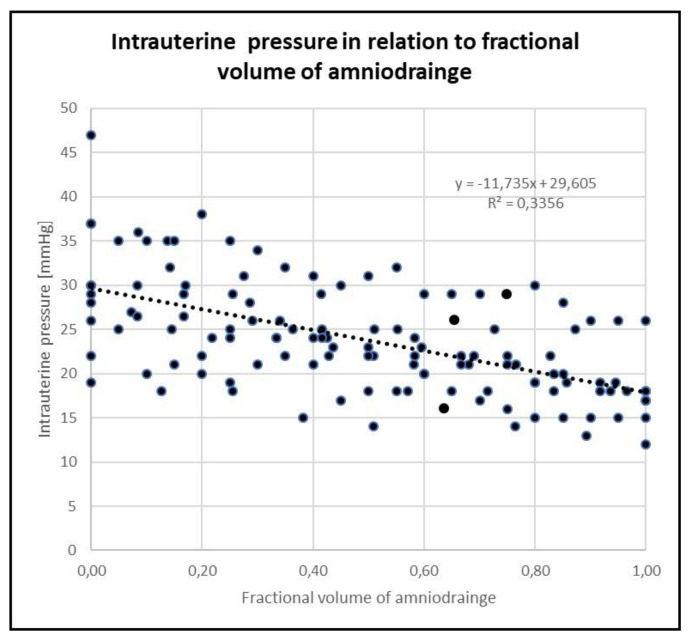
Intrauterine pressure (mmHg) in relation to fractional amniodrainage volume (0 to 1) following FLA. Data on intrauterine pressures (y) are presented as a function of fractional amniodrainage volume (x). The linear regression (broken line) is given as y = -11.7x + 29.6; R2 = 0.34, n = 10. Fractional amniodrainage volume is defined as drained volume at time point of measurement divided by total amniodrainage volume. Measurements were performed every 200 mL of amniodrainage. FLA: fetoscopic laser ablation.

**Figure 2 jcm-09-02085-f002:**
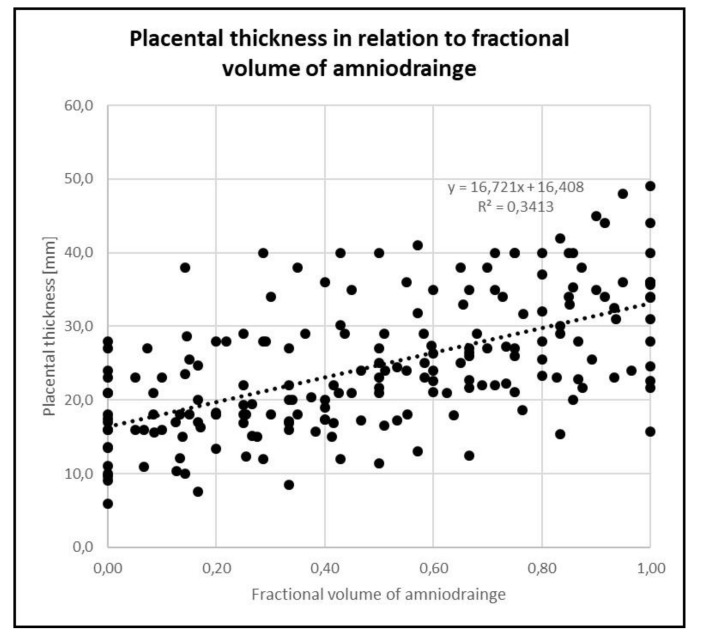
Placental thickness (mm) in relation to fractional amniodrainage volume (0 to 1) following FLA. Data on placental thickness (y) are presented as a function of fractional amniodrainage volume (x). The linear regression (broken line) is given as y = 16.7x + 16.4; R² = 0.34, n = 18. Fractional amniodrainage volume is defined as drained volume at time point of measurement divided by total amniodrainage volume. Measurements were performed every 200 mL of amniodrainage. FLA: fetoscopic laser ablation.

**Table 1 jcm-09-02085-t001:** Patient characteristics and Quintero stage at intervention.

N *	18 (100)
Maternal age, years ^§^	28 (21–39)
Gestational age at intervention, weeks ^§^	21.1 (19.3–28.1)
BMI ^§^	25.9 (17.4–38.9)
Quintero stage at intervention *	
Stage I	6 (33.3)
Stage II	5 (27.8)
Stage III	7 (39.9)
Stage IV	0 (0)
Stage V	0 (0)

* Data are presented as numbers (%); ^§^ Data are presented as medians (range).

**Table 2 jcm-09-02085-t002:** Perioperative maternal blood characteristics and intraoperative measurements before and after amniodrainage.

	Before	After	*p*-Value
Hemoglobin (n = 18) *	112.1 ± 12.0 g/L	95.1 ± 12.1 g/L	<0.001
Hematocrit (n = 18) *	33.38 ± 3.77 [%]	28.41 ± 3.50 [%]	<0.001
Serum albumin (n = 7) *	37.9 ± 0.90 g/L	30.7 ± 2.21 g/L	<0.001
Intrauterine pressure (n = 10) *	30.05 ± 8.10 mmHg	17.60 ± 3.56 mmHg	<0.001
Relative change in intrauterine pressure (n = 10) ^§^	100 ± 0.00 %	59 ± 7.66 %	<0.001
Placental thickness (n = 18) *	16.76 ± 6.35 mm	31.83 ± 8.64 mm	<0.001
Relative change in placental thickness (n = 18) ^§^	100 ± 0.00 %	205 ± 47.00 %	<0.001

* Data are presented as mean values (SD); § Data are presented as percentages (SD).

**Table 3 jcm-09-02085-t003:** Maternal hemodilution following amniodrainage.

Hematologic Parameter *	Relative Changes from pre to Post Intervention	*p*-Value
DB-Hb (n = 18)	0.85 ± 0.07	<0.001
DB-Hct (n = 18)	0.85 ± 0.08	<0.001
DB-Hb (n = 7)	0.85 ± 0.04	<0.001
DB-Hct (n = 7)	0.85 ± 0.05	<0.001
DP (n = 7)	0.81 ± 0.05	<0.001
RBV-Hb (n = 7)	1.17 ± 0.06	<0.001
RBV-Hct (n = 7)	1.18 ± 0.07	<0.001
RPV (n = 7)	1.24 ± 0.08	<0.001

* Mean relative hemodilution for blood (DB) as relative changes in hemoglobin (DB-Hb) and hematocrit (DB-Hct) and plasma protein (DP) displayed as the ratio of post to pre concentrations. Relative expansion of blood (RBV) and plasma volumes (RPV) by hemodilution was quantified as the ratio of pre to post concentrations. Calculations of maternal plasma volume changes, based on serum albumin measurements (n = 7), were representative of the entire study population (n = 18). Data are presented as mean values (SD).

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
