# Peer review of "Amniodrainage-Induced Circulatory Dysfunction in Women Treated for Twin-To-Twin Transfusion Syndrome"

_jcm, 2020, doi:10.3390/jcm9072085_

Round 1

Reviewer 1 Report

To authors,

The authors showed that amniodrainage at the time of FLP for TTTS led to uterine decompression, increased placental thickness and subsequent maternal hemodilution. The authors, considering the resemblance of this phenomenon to mirror syndrome, refer this phenomenon to “amniodrainage-induced circulatory dysfunction (AICD)”. I have some suggestions.

  1. The amniodrainage was done at various gestational stage/week. Naturally, plasma volume expansion (observed in normal pregnancy) greatly differs according to the gestational age. You related data with only “reduction volume” and not pay attention to gestational week (and thus original plasma expansion state according to gestational age). Please touch this issue. Please confirm 1) if this is OK (then state so), 2) this is the limitation (then state so), or 3) you may be able to provide another data (taking gestational week into account) (then, provide this data). Please take action; 1), 2) or 3).
  2. You pointed that there may be resemblance between the present phenomenon and mirror syndrome, which is very attractive. Then, the present hemodilution is “bad effect”, isn’t it? You provided a new nomenclature of AICD; “D” (dysfunction) is OK? In mirror syndrome, hemodilution may cause lung edema or cardiac failure (overdistention) and thus this hemodilution may be “D” (bad effect). Then, this also holds true to the present phenomenon (hemodilution)? Please shortly state your view.
  3. You stated three possible reasons why hemodilution occurs. But you did not “examine/confirm” any of them in this study. You are not blamed for this; however, you had better touch this issue (very shortly); for example, “Although we provided some possible mechanisms of the present phenomenon, we did not examine/confirm if these really occur in these patients.” (example).
  4. You have already provided the concept of the present phenomenon in Reference 5. In my opinion, you had better state definitely “what was examined and identified in reference 5 study” and then, “what was further added in this study”. In short, in Introduction, you cited reference 5 in a different manner but you had better definitely state what was done in reference 5 study. Anyhow, it is evident that the present study was the “confirmation” or “detailed observation” of the data of reference 5. Now that the paper writing ethics is stringent (in my opinion, “too” stringent), you had better be much cautious for this issue. Please slightly touch this point. No need to write long. Please do not make the manuscript any longer. The present one is concise and I love this conciseness.

Reviewer 2 Report

A well-constructed paper with an interesting subject.

I would suggest adding information about the pulsatility index in the middle cerebral artery before and after amniodrainage as it would be of interest to many readers and would add impact to the manuscript.

The only issue is the small sample size.
